# The prognostic value of thromboelastography MA/R ratio in predicting mortality in acute respiratory failure patients

Zhang-Sheng Zhao[1]*, Zhen-Zhen Wang[1], Lei Wang[1], Li-Hui Qian[1], Bin Hu[2], You-Li Ma[1]

**1** Transfusion Medicine Center, Apheresis and Transfusion Therapy Center, Ningbo Medical Center Lihuili Hospital, Ningbo, Zhejiang, People's Republic of China, **2** Department of Clinical Laboratory, Ningbo Medical Center Lihuili Hospital, Ningbo, Zhejiang, People's Republic of China

* zszhaonb@163.com

## Abstract

Thromboelastography (TEG) MA/R ratio reflects coagulation status and thrombus strength. This study evaluated its prognostic value in acute respiratory failure (ARF). A retrospective analysis of 371 ARF patients admitted to the ICU, stratified by MA/R quartiles. Outcomes included 28-day mortality, deep vein thrombosis (DVT), mechanical ventilation duration, and ICU stay. Cox proportional hazards regression model was used to assess hazard ratios, restricted cubic spline was employed to evaluate the nonlinear relationship between MA/R and mortality, and Kaplan-Meier analysis was conducted to compare survival time across different MA/R groups. Patients in the lowest MA/R quartile (Q1) had significantly higher 28-day mortality (59.8% vs. 22.1–28.0% in Q2-Q4; $P < 0.001$) and elevated inflammatory markers (cytokines, procalcitonin, lactate, creatinine; $P < 0.05$). DVT incidence, ventilation duration, and ICU stay did not differ between groups. Multivariate analysis identified MA/R as an independent mortality predictor ($P < 0.05$), with mortality risk sharply increasing below a threshold of 9.7. Kaplan-Meier curves showed shorter survival in Q1 ($P < 0.001$). The MA/R ratio measured at ICU admission can rapidly identify coagulation dysfunction in patients with acute respiratory failure, with a low MA/R ratio being a strong indicator of poor prognosis.

## Introduction

Acute respiratory failure (ARF) is a critical condition characterized by the inability of the respiratory system to maintain adequate gas exchange, with high morbidity and mortality rates [1]. The mortality rate for ARF remains alarmingly high, typically ranging from 30% to 60% [2], and may be even higher in certain patient populations, particularly those in the intensive care unit (ICU) [3].

ARF is often associated with a systemic inflammatory response, which significantly impacts coagulation function. The systemic inflammatory response, driven by

**Data availability statement:** All relevant data are within the paper and its Supporting Information files.

**Funding:** This work was supported by the Green Kou Foundation of the Zhejiang Blood Transfusion Association (ZJB-LK-2023-006, Zhang-Sheng Zhao), the Medical and Health Science and Technology Project of Zhejiang Province (2020KY859, Lei Wang), and the Luili Foundation of Lihuili Hospital (2022YB004, Zhang-Sheng Zhao).

**Competing interests:** The authors have declared that no competing interests exist.

pro-inflammatory cytokines (e.g., IL-6, TNF-α), can activate the coagulation system [4]. Inflammation not only directly triggers coagulation factors but also inhibits natural anticoagulant mechanisms (e.g., protein C system, antithrombin), leading to a prothrombotic state [5].

Monitoring coagulation parameters such as activated partial thromboplastin time (APTT), prothrombin time (PT), D-dimer, and fibrinogen in ARF patients reflects disease severity and provides valuable insights for clinical management and treatment strategies [6]. However, traditional coagulation markers are not superior to widely used critical illness scoring systems, such as the Sequential Organ Failure Assessment (SOFA) and Acute Physiology and Chronic Health Evaluation II (APACHE II), in predicting the outcomes of acute respiratory failure [7].

Thromboelastography (TEG) offers significant advantages over conventional coagulation tests, providing real-time, comprehensive assessment of hemostasis [8]. This capability enables a more detailed understanding of a patient's hemostatic status, making TEG widely applicable in critically ill patients with trauma [9], sepsis [10], and disseminated intravascular coagulation (DIC) [11]. Although TEG has been used in intensive care settings and can predict mortality in critically ill patients, there is limited research examining the relationship between TEG and the prognosis of respiratory failure. Additionally, the MA/R ratio has been recognized as a rapid and comprehensive indicator that reflects early coagulation and thrombus strength, with significant advantages in identifying and predicting mortality in traumatic coagulopathies [12]. Based on this, we hypothesize that the MA/R ratio, measured upon ICU admission, can quickly identify coagulation dysfunction in ARF patients and offer greater clinical relevance in predicting the prognosis of acute respiratory failure when compared to traditional markers.

## Methods

### Study setting

This retrospective study included 476 patients who were admitted to the intensive care unit (ICU) of Ningbo Li Huili Medical Center due to acute respiratory failure between 01/01/ 2021 and 31/05/2024. Data Access Dates: The patient data were accessed for research purposes on 11/15/2024. This study was conducted in accordance with the ethical principles of the Declaration of Helsinki and was approved by the Ethics Committee of Ningbo Medical Center Lihuili Hospital (Approval No. KY2024SL449−01). Due to the retrospective nature of the study, the Ethics Committee of Li Huili Hospital waived the requirement for obtaining informed consent.

Inclusion criteria: 1) Patients who met the diagnostic criteria for acute respiratory failure and required assisted ventilation therapy were admitted to the ICU. 2) Thromboelastography (TEG) testing was performed on the day of ICU admission. Exclusion criteria: The following patients were excluded: 1) those with an ICU stay of less than 48 hours (n = 30); 2) those with incomplete clinical data (n = 24); 3) patients with repeated ICU admissions (n = 11); 4) patients who received plasma, cryoprecipitate, or platelet transfusion before ICU admission (n = 9). 5) Patients with active bleeding (n = 10). 6) Patients with neurogenic respiratory failure (n = 15). 7) Patients with refractory malignant arrhythmias (n = 6).

For patients undergoing heparin anticoagulation therapy, heparinase was used to neutralize heparin and correct TEG results. Although some patients were receiving antiplatelet therapy (e.g., aspirin or clopidogrel), which could potentially alter platelet aggregation in vitro, these two drugs do not affect kaolin-induced TEG results [13]. Therefore, this population was also included in the analysis. Patients were divided into four groups based on the quartiles of their MA/R ratio: MA/R1 (0%−25%), MA/R2 (25%−50%), MA/R3 (50%−75%), and MA/R4 (75%−100%). This quartile-based classification was adopted to ensure balanced subgroup sizes and to explore potential outcome differences across the MA/R strata. The rationale for using four categories follows the approach proposed by Savage et al. [12], in which quartile stratification of the MA/R ratio was shown to effectively differentiate mortality risk in trauma patients. The flowchart of this study is shown in Fig 1.

## Study variables and clinical outcomes

Patient data, including age, sex, comorbidities (e.g., pneumonia, sepsis, malignancy, cerebrovascular/cardiovascular diseases, renal/liver insufficiency, trauma, diabetes, hypertension), vital signs (e.g., temperature, heart rate, mean arterial pressure, respiratory rate), disease severity scores (e.g., SOFA score, APACHE II score), and laboratory markers (e.g., white

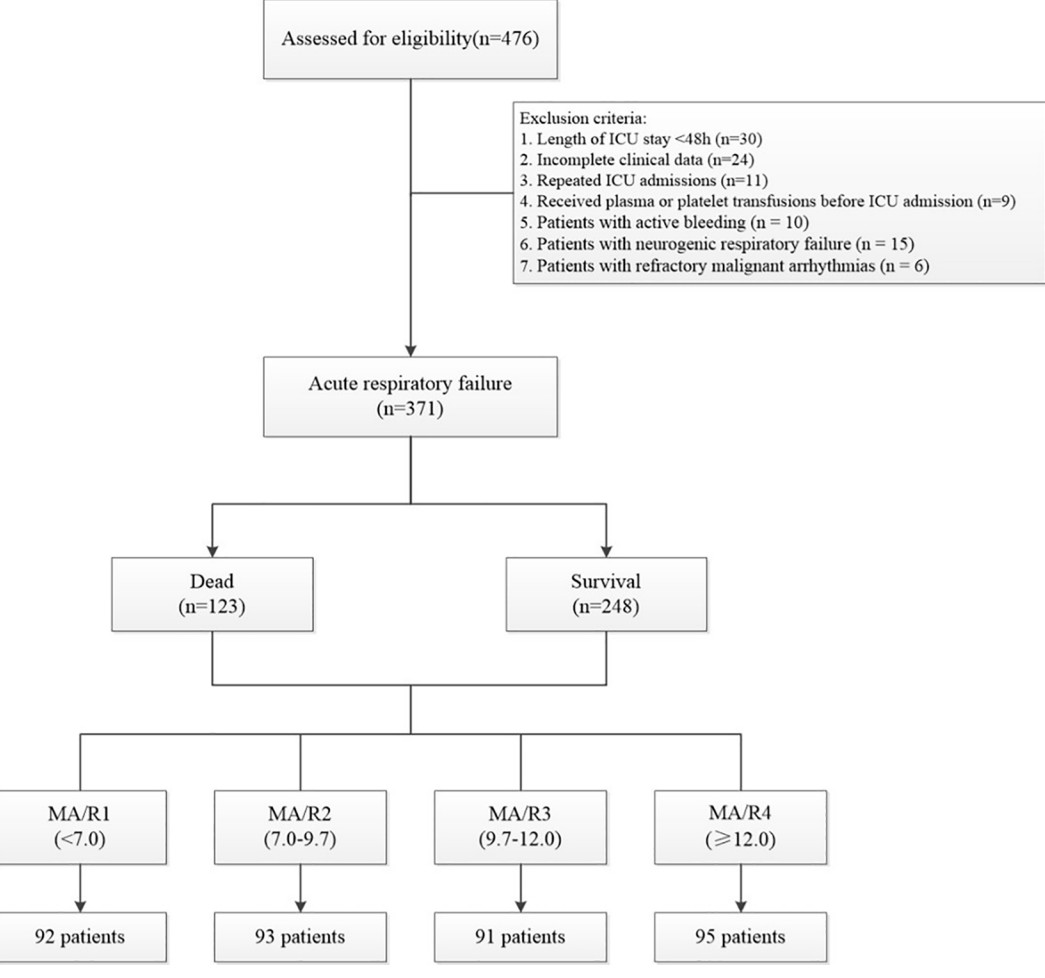

**Fig1. Flow chart of the study.**

blood cell count, hemoglobin, platelet count, PT, APTT, TT, fibrinogen, D-dimer, lactate, CRP, PCT, IL-6, IL-10, albumin), were collected from the electronic medical records system. The primary outcome of this study was the 28-day mortality rate. Secondary outcomes included the incidence of deep vein thrombosis, duration of mechanical ventilation, and ICU length of stay.

### Thromboelastography (TEG) measurement

TEG parameters in this study were generated using the Thromboelastograph® 5000 Hemostasis System (Haemoscope, USA). The process involved preparing the analyzer and reagents, followed by blood collection in a sodium citrate tube. After warming the reagents to 37°C, 1 mL of citrated blood was transferred to a kaolin vial. Then, 340 µL of the activated blood was placed into the TEG cup, which was pre-filled with 20 µL of $CaCl_2$. The cup and pin were securely attached to the analyzer. The appropriate test type was selected, patient information was entered, and the test was initiated. The test duration was between 30 and 60 minutes. Key parameters, including reaction time (R), clot kinetics time (K), alpha angle (α), and maximum amplitude (MA), were recorded. The MA/R ratio was calculated by dividing the MA by R. The interpretation of TEG parameters and reference intervals are detailed in S1 Table.

### Statistical analyses

Statistical analysis and data processing were performed using SPSS (IBM, 26.0, USA), RStudio (Posit PBC, 2024.12.1, USA), and GraphPad Prism software (GraphPad, 8.0, USA). Continuous variables are presented as median (interquartile range), while categorical variables are expressed as n (percentage). Intergroup comparisons of variables were conducted using Chi-square tests, Mann-Whitney U-test, or the Kruskal-Wallis test, as appropriate. Linear regression was applied to diagnose collinearity, and variables with a variance inflation factor (VIF) greater than 10 were excluded. The Cox proportional hazards model was used to assess the hazard ratios of different MA/R ratios on survival time. The restricted cubic splines (RCS) were employed to evaluate the nonlinear relationship between MA/R and mortality. Kaplan-Meier survival analysis was employed to evaluate survival differences among groups with varying MA/R ratios, and the log-rank test was used for comparison. Statistical significance was defined as $P < 0.05$, with two-tailed tests.

## Results

### Baseline characteristics of the study population

A total of 476 acute respiratory failure patients admitted to the ICU and undergoing TEG testing were enrolled in this study. After applying exclusion criteria, 371 patients were included in the final analysis. Among them, 250 were male and 121 were female, with a median age of 73 (interquartile range: 63, 81) years. The 28-day mortality rate for acute respiratory failure patients was notably high at 33.2%. Compared to the survival group, the non-survival group had a higher proportion of male patients, a higher incidence of sepsis, more frequent renal and hepatic dysfunction, lower mean arterial pressure (MAP), and higher APACHE II scores ($P < 0.05$, Table 1).

In terms of laboratory parameters, the non-survival group exhibited significantly lower hemoglobin levels, lower platelet counts, prolonged prothrombin time (PT) and activated partial thromboplastin time (APTT), higher fibrinogen and D-dimer levels, elevated lactate levels, higher procalcitonin (PCT) levels, elevated cytokine levels (IL-6 and IL-10), increased creatinine levels, and lower albumin levels compared to the survival group ($P < 0.05$, Table 1). Regarding clinical outcomes, the non-survival group had a longer duration of mechanical ventilation than the survival group ($P = 0.03$, Table 1). However, no significant differences were observed between the groups for other clinical outcomes, such as deep vein thrombosis (VTE) and ICU length of stay (LOS) ($P > 0.05$, Table 1).

### TEG parameters between survival and non-survival groups

TEG parameters showed statistically significant differences between the survival and non-survival groups ($P < 0.05$, Table 1). In patients with acute respiratory failure (ARF), the non-survival group exhibited prolonged R and K values

**Table 1. Baseline characteristics of the study population based on 28-day survival status.**

| Variables | Total (n = 371) | non-Survival (n = 123) | Survival (n = 248) | P |
|---|---|---|---|---|
| Sex(male) (n, %) | 250(67.4) | 92(74.8) | 158(63.7) | **0.032** |
| Age (years) | 73(63, 81) | 73(63, 84) | 73(63, 80) | 0.336 |
| Comorbidities (n, %) | | | | |
| Pneumonia | 343(92.5) | 111(90.2) | 232(93.5) | 0.257 |
| Sepsis | 139(37.5) | 58(47.2) | 81(32.7) | **0.007** |
| Tumor | 79(21.3) | 27(22.0) | 52(21.0) | 0.828 |
| Cerebrovascular disease | 193(52.0) | 60(48.8) | 133(53.6) | 0.379 |
| Cardiovascular disease | 217(58.5) | 69(56.1) | 148(59.7) | 0.510 |
| Renal dysfunction | 86(23.2) | 49(39.8) | 37(14.9) | **<0.001** |
| Hepatic dysfunction | 22(5.9) | 15(12.2) | 7(2.8) | **<0.001** |
| Diabetes | 112(30.2) | 34(27.6) | 78(31.5) | 0.452 |
| Hypertension | 182(49.1) | 58(47.2) | 124(50.0) | 0.606 |
| Vital signs | | | | |
| Body temperature (°C) | 36.8(36.4,37.5) | 36.8(36.3, 37.6) | 37.0(36.5, 37.3) | 0.357 |
| Heart rate (bpm) | 93(83, 109) | 98(85, 110) | 91(83, 109) | 0.421 |
| MAP (mmHg) | 87(75, 101) | 83(70, 95) | 90(78, 102) | **0.001** |
| Respiratory rate (bpm) | 18 (15 22 ) | 18(16, 24) | 18(15, 22) | 0.179 |
| Severity scores | | | | |
| SOFA score | 6 (4 10 ) | 7 (4 10 ) | 6 (4 9) | 0.063 |
| APACHE II score | 16 (11, 23) | 21 (12, 25) | 16(12, 21) | **0.024** |
| Laboratory parameters | | | | |
| Leukocytes (×10⁹/L) | 11.4(7.5, 15.3) | 12.3(7.7, 16.3) | 10.9(7.4, 15.0) | 0.553 |
| Hemoglobin (g/L) | 106(88, 120) | 98(76, 112) | 107(85, 121) | **0.010** |
| Platelets (×10⁹/L) | 164(97, 223) | 136(94, 196) | 165(100, 216) | **<0.001** |
| PT (s) | 14.7(13.2, 16.9) | 15.3(13.4, 19.6) | 14.8(13.2, 16.8) | **0.003** |
| APTT (s) | 37.5(31.0, 47.3) | 39.0(29.1, 54.7) | 36.4(30.7, 46.8) | **0.006** |
| TT (s) | 16.9(15.5, 18.8) | 16.9(15.1, 20.9) | 16.9(15.5, 18.2) | 0.090 |
| Fibrinogen (g/L) | 3.99(2.75, 5.41) | 4.07(1.70, 5.33) | 3.88(2.89, 5.01) | **0.016** |
| D-dimer (mg/L) | 2.13(1.14, 6.48) | 4.45(1.93, 8.71) | 2.27(1.20, 6.47) | **0.008** |
| Lactate (mmol/L) | 1.70(1.10, 3.45) | 1.70(1.10, 4.20) | 1.55(1.00, 3.00) | **0.026** |
| CRP (mg/L) | 71.6(22.5, 162.2) | 97.6(43.9, 190.2) | 76.4(19.7, 140.1) | 0.053 |
| PCT (ng/mL) | 1.01(0.28, 3.37) | 1.74(0.58, 8.01) | 0.81(0.26, 2.49) | **<0.001** |
| IL-6 (pg/mL) | 114.8(32.4, 466.1) | 164.2(63.2, 624.8) | 113.3(31.0, 490.9) | **0.002** |
| IL-10 (pg/mL) | 13.1(7.1, 39.8) | 22.8(9.4, 46.7) | 12.7(6.9, 46.4) | **0.029** |
| Albumin (g/L) | 29.5(26.4, 33.1) | 28.4(23.3, 30.6) | 29.0(25.7, 32.9) | **0.040** |
| Creatinine (μmol/L) | 91.0(57.5, 160.0) | 115.0(77.0, 174.0) | 77.5(57.0, 139.8) | **<0.001** |
| TEG pameters | | | | |
| R (min) | 6.6(5.4, 8.4) | 7.3(5.6, 10.2) | 6.3(5.4, 7.6) | **<0.001** |
| K (min) | 1.8(1.3, 2.7) | 2.0(1.2, 3.7) | 1.7(1.3, 2.1) | **<0.001** |
| α-angle (deg) | 65.4(56.1, 70.9) | 63.1(47.5, 72.8) | 66.3(60.8, 70.8) | **<0.001** |
| MA (mm) | 64.5(56.1, 64.5) | 59(48.3, 70.4) | 64.6(58.4, 71.6) | **0.001** |
| MA/R (mm/min) | 9.7(7.0, 12.0) | 8.8(5.0, 11.7) | 10.1(8.3, 12.1) | **<0.001** |
| Outcomes | | | | |
| DVT (%) | 52(14.0) | 14(11.4) | 38(15.3) | 0.303 |
| Mechanical ventilation (days) | 8(4, 19) | 10(5, 22) | 7(3, 17) | **0.003** |
| ICU LOS (days) | 14(7, 27) | 16(8, 28) | 13(7, 25) | 0.087 |

*MA/R* MA/R ratio, *MAP* Mean arterial pressure, *PT* Prothrombin time, *APTT* Activated partial thromboplastin time, *TT* Thrombin time, *CRP* C-reactive protein, *PCT* Procalcitonin, *IL* Interleukin, *R* Reaction time, *K* Clot kinetics time, *MA* Maximum amplitude, *DVT* Deep venous thrombosis, *LOS* Length of stay.

and decreased α-angle, and MA compared with the survival group. The median MA/R ratio was significantly lower in non-survivors (8.8 [5.0–11.7]) than in survivors (10.1 [8.3–12.1], $P<0.001$). A lower MA/R ratio indicates a slower process of clot development and weaker clot strength relative to the initial coagulation phase, suggesting a hypocoagulable or less efficient coagulation state in non-survivors. Although these differences were statistically significant, most patients' TEG parameters remained within or near the reference range. To further investigate the prognostic implications of coagulation balance, patients were divided into four subgroups according to the quartiles of their MA/R ratio: MA/R1 group (MA/R < 7.0, n = 92), MA/R2 group (7.0 ≤ MA/R < 9.7, n = 93), MA/R3 group (9.7 ≤ MA/R < 12, n = 91), and MA/R4 group (MA/R ≥ 12, n = 95).

## Association between MA/R ratio and 28-day mortality

We found that the MA/R1 group was more likely to experience hepatic and renal dysfunction, lower MAP, and higher respiratory rates compared to the other quartile groups. In terms of laboratory parameters, the MA/R1 group exhibited more abnormal values in platelet counts, coagulation indicators (PT, APTT, TT, fibrinogen, D-dimer), lactate, PCT, cytokines (IL-6 and IL-10), creatinine, and albumin compared to the other quartile groups ($P<0.05$, Table 2). Additionally, the TEG parameters in the MA/R1 group demonstrated more extreme values compared to the other quartile groups ($P<0.05$, Table 2), with significantly prolonged R and K values, significantly reduced α-angle and MA values, and most indicators exceeding the normal range. Notably, the 28-day mortality rate in the MA/R1 group was significantly higher than in the other groups (59.8%, 28.0%, 23.1%, and 22.1%, respectively; $P<0.001$, Table 2).

## Multivariate analysis of factors associated with mortality

To further analyze the hazard ratios of various variables contributing to mortality in patients with acute respiratory failure, we included the parameters with statistical significance from Table 1 in a univariate Cox regression analysis. Continuous variables were categorized using the median value as the cutoff. A collinearity diagnosis of the variables was conducted, and the K and α-angle indicators were excluded due to their variance inflation factors (VIF) exceeding 10. The univariate Cox regression analysis revealed that comorbid renal (Yes vs. No, HR = 0.535, $P=0.001$) and hepatic (Yes vs. No, HR = 0.439, $P=0.003$) dysfunction, MAP (Low vs. High, HR = 0.613, $P=0.008$), APACHE II scores (Low vs. High, HR = 1.610, $P=0.031$), platelet count (Low vs. High, HR = 0.642, $P=0.018$), PCT (Low vs. High, HR = 1.681, $P=0.006$), IL-6 (Low vs. High, HR = 1.638, $P=0.016$), creatinine (Low vs. High, HR = 1.872, $P=0.001$), R (Low vs. High, HR = 1.841, $P=0.001$), MA (Low vs. High, HR = 0.664, $P=0.027$), and MA/R ratio (MA/R1 vs. MA/R2, HR = 0.348, $P<0.001$; MA/R1 vs. MA/R3, HR = 0.284, $P<0.001$; MA/R1 vs. MA/R4, HR = 0.323, $P<0.001$) were closely associated with mortality in patients with acute respiratory failure (Table 3).

  When these indicators were included in a multivariate regression analysis (Backward LR), the final model identified that MAP (Low vs. High, HR = 0.490, $P=0.004$), APACHE II score (Low vs. High, HR = 1.558, $P=0.070$), creatinine (Low vs. High, HR = 1.704, $P=0.036$), and MA/R ratio (MA/R1 vs. MA/R2, HR = 0.439, $P=0.007$; MA/R1 vs. MA/R3, HR = 0.272, $P<0.001$; MA/R1 vs. MA/R4, HR = 0.393, $P=0.003$) were closely associated with mortality in patients with acute respiratory failure (Table 3).

## Predictive value of MA/R ratio

Restricted cubic spline curves demonstrated a U-shaped association between the MA/R ratio and 28-day mortality. Mortality risk increased rapidly with decreasing MA/R ratio when <9.7, and began to rise gradually when the MA/R ratio exceeded 15.7 (Fig 2). Kaplan-Meier survival analysis demonstrated that the survival time in the MA/R1 group was significantly shorter than in the other MA/R quartile groups ($P<0.001$); the survival time of the high creatinine group was significantly shorter than that of the low creatinine group ($P<0.001$); the survival time of the low MAP group was significantly shorter than that of the high MAP group ($P=0.004$); and the survival time of the high APACHE II score group was significantly shorter than that of the low APACHE II score group ($P=0.003$) (Fig 3).

 

**Table 2. Characteristics of studied patients divided into MA/R ratio Groups by interquartile range.**

| Variable | MA/R1 | MA/R2 | MA/R3 | MA/R4 | P |
|---|---|---|---|---|---|
| | (n = 92) | (n = 93) | (n = 91) | (n = 95) | |
| Sex(male) (n, %) | 65(70.7) | 61(65.6) | 68(74.7) | 56(58.9) | 0.117 |
| Age (years) | 72(62, 81) | 75(63, 82) | 70(61, 78) | 76(69, 84) | **0.017** |
| Comorbidities (n, %) | | | | | |
| Pneumonia | 83(90.2) | 87(93.5) | 84(92.3) | 89(93.7) | 0.796 |
| Sepsis | 38(41.3) | 35(37.6) | 37(40.7) | 29(30.5) | 0.403 |
| Tumor | 25(27.2) | 16(17.2) | 23(25.3) | 15(15.8) | 0.144 |
| Cerebrovascular disease | 41(44.6) | 50(53.8) | 48(52.7) | 54(56.8) | 0.382 |
| Cardiovascular disease | 52(56.2) | 58(62.4) | 57(62.6) | 50(52.6) | 0.439 |
| Renal dysfunction | 36(39.1) | 20(21.5) | 20(22.0) | 10(10.5) | **< 0.001** |
| Hepatic dysfunction | 10(10.9) | 6(6.5) | 6(6.6) | 0(0.0) | **0.017** |
| Diabetes | 24(26.1) | 30(32.3) | 26(28.6) | 32(33.7) | 0.662 |
| Hypertension | 36(39.1) | 48(51.6) | 45(49.5) | 53(55.8) | 0.133 |
| Vital sign | | | | | |
| Body temperature (°C) | 36.7(36.2, 37.4) | 36.8(36.5,37.5) | 36.9(36.3,37.4) | 36.9(36.5,37.6) | 0.262 |
| Heart rate (bpm) | 99(86, 114) | 94(82, 111) | 90(80,105) | 93(79,107) | 0.096 |
| MAP (mmHg) | 82(71, 100) | 87(73, 102) | 89(79, 102) | 90(77, 100) | **0.001** |
| Respiratory rate (bpm) | 19(16, 23) | 18(15,20) | 17(15, 20) | 18(15, 22) | **0.013** |
| Severity scores | | | | | |
| SOFA score | 8(4, 11) | 7(4, 10) | 5(4, 9) | 6(4, 9) | **0.031** |
| APACHE II score | 20(9, 27) | 17(13, 25) | 14(11, 21) | 16(11,22) | 0.067 |
| Laboratory parameters | | | | | |
| Leukocytes (×10⁹/L) | 11.6(7.2, 15.3) | 11.7(7.2, 14.6) | 10.5(7.0,15.1) | 12.3(8.3, 15.9) | 0.053 |
| Hemoglobin (g/L) | 103(76, 121) | 103(91, 122) | 108(92, 120) | 102(90, 117) | 0.124 |
| Platelets (×10⁹/L) | 92(56,179) | 156(97, 211) | 173(104, 222) | 197(149, 378) | **<0.001** |
| PT (s) | 17.2(14.1, 23.2) | 14.8(13.6, 16.9) | 14.5(13.0, 15.5) | 14.2(12.8, 15.8) | **<0.001** |
| APTT (s) | 47.6(36.0, 66.2) | 39.9(33.3, 49.9) | 35.8(31.1, 42.8) | 30.7(27.8, 38.9) | **<0.001** |
| TT (s) | 18.2(16.4, 26.1) | 17.3(15.8, 20.0) | 16.4(15.0, 18.1) | 16.2(14.7, 17.3) | **<0.001** |
| Fibrinogen (g/L) | 3.05(1.83, 4.68) | 4.36(2.74, 5.88) | 4.09(2.96, 5.53) | 4.2(3.15, 5.37) | **<0.001** |
| D-dimer (mg/L) | 3.99(1.27, 11.2) | 2.41(1.12, 8.59) | 1.59(1.01, 5.10) | 1.95(1.14, 5.41) | **0.009** |
| Lactate (mmol/L) | 2.90(1.50, 6.15) | 1.80(1.20, 3.70) | 1.40(1.00, 2.30) | 1.40(0.90, 2.40) | **<0.001** |
| CRP (mg/L) | 75.9(27.5, 168.5) | 80.4(23.7, 183.1) | 68.8(16.6, 159.5) | 52.8(16.3, 131.0) | 0.568 |
| PCT (ng/mL) | 3.32(0.60, 18.35) | 1.22(0.40, 2.82) | 0.60(0.15, 3.00) | 0.50(0.20, 1.87) | **<0.001** |
| IL-6 (pg/mL) | 840.6(130.5, 2500.0) | 121(28.3, 417.5) | 54.3(21.8, 236.2) | 63.7(17.4, 162.8) | **<0.001** |
| IL-10 (pg/mL) | 48.6(14.3, 203.0) | 14.1(7.67, 34.0) | 9.4(6.2, 22.8) | 7.3(3.4, 14.8) | **<0.001** |
| Albumin (g/L) | 28.6(24.2, 31.9) | 29.3(26.2, 33.1) | 29.5(26.9, 33.4) | 29.8(27.0, 33.0) | **0.038** |
| Creatinine (μmol/L) | 133(81, 213) | 101(61, 170) | 70(48, 121) | 77(56, 126) | **<0.001** |
| TEG pameters | | | | | |
| R (min) | 11.3(9.2, 17.2) | 7.4(6.7, 8.2) | 5.8(5.5, 6.6) | 4.8(4.2, 5.2) | **<0.001** |
| K (min) | 3.8(2.7, 6.0) | 1.8(1.6, 2.2) | 1.6(1.3, 2.1) | 1.2(1.1, 1.5) | **<0.001** |
| α-angle (deg) | 46.4(35.7, 55.4) | 65.1(60.5, 68.1) | 67.1(62.4, 70.8) | 72.5(69.6, 75.4) | **<0.001** |
| MA (mm) | 53.3(41.6, 63.2) | 65.2(58.3, 70.5) | 64.7(58.8,70.1) | 69.9(64.5, 73.5) | **<0.001** |
| Outcomes | | | | | |
| DVT (%) | 12(13.0) | 13(14.0) | 13(14.3) | 14(14.7) | 0.989 |
| Mechanical ventilation (days) | 8(4, 16) | 10(4, 20) | 8(4, 24) | 8(4,16) | 0.769 |
| ICU LOS (days) | 11(5, 22) | 17(8, 28) | 17(7, 28) | 14(8, 28) | 0.051 |
| Mortality (28 days) | 55(59.8) | 26(28.0) | 21(23.1) | 21(22.1) | **<0.001** |

*MA/R1* First quartile group of the MA/R ratio, *MA/R2* Second quartile group of the MA/R ratio, *MA/R3* Third quartile group of the MA/R ratio, *MA/R4* Fourth quartile group of the MA/R ratio, *MAP* Mean arterial pressure, *PT* Prothrombin time, *APTT* Activated partial thromboplastin time, *TT* Thrombin time, *CRP* C-reactive protein, *PCT* Procalcitonin, *IL* Interleukin, *R* Reaction time, *K* Clot kinetics time, *MA* Maximum amplitude, *DVT* Deep venous thrombosis, *LOS* Length of stay.

**Table 3. Univariate and multivariate regression models for survival in the study population.**

| Variables | Univariate Analysis | | Multivariate Analysis | |
|---|---|---|---|---|
| | HR (95% CI) | *P* | HR (95% CI) | *P* |
| Sex | 0.749 (0.498, 1.126) | 0.165 | / | / |
| Sepsis | 0.873 (0.611, 1.248) | 0.458 | / | / |
| Renal dysfunction | 0.535 (0.372, 0.769) | **0.001** | / | **/** |
| Hepatic dysfunction | 0.439 (0.255, 0.753) | **0.003** | / | / |
| MAP | 0.613 (0.427, 0.881) | **0.008** | 0.490 (0.302, 0.794) | **0.004** |
| APACHE II | 1.610 (1.046, 2.480) | **0.031** | 1.558 (0.964, 2.518) | 0.070 |
| Hemoglobin | 0.876(0.610, 1.258) | 0.474 | / | / |
| Platelets | 0.642 (0.445, 0.925) | **0.018** | / | / |
| PT | 1.369(0.954, 1.964) | 0.088 | / | / |
| APTT | 1.352 (0.946, 1.932) | 0.098 | / | / |
| Fibrinogen | 0.728 (0.509, 1.043) | 0.084 | / | / |
| D-dimer | 1.404(0.976, 2.020) | 0.067 | / | / |
| Lactate | 1.271 (0.867, 1.863) | 0.220 | / | / |
| PCT | 1.681 (1.163, 2.429) | **0.006** | / | / |
| IL-6 | 1.638 (1.095, 2.451) | **0.016** | / | / |
| IL-10 | 1.435 (0.961, 2.142) | 0.077 | / | / |
| Albumin | 0.871 (0.609, 1.244) | 0.446 | / | / |
| Creatinine | 1.872 (1.291, 2.714) | **0.001** | 1.704 (1.035, 2.803) | **0.036** |
| R | 1.841 (1.277, 2.656) | **0.001** | / | / |
| MA | 0.664 (0.462, 0.954) | **0.027** | / | / |
| MA/R | | | | |
| MA/R1vsMA/R2 | 0.348 (0.218, 0.556) | **<0.001** | 0.439 (0.243, 0.795) | **0.007** |
| MA/R1vsMA/R3 | 0.284 (0.171, 0.470) | **<0.001** | 0.272 (0.139, 0.532) | **<0.001** |
| MA/R1vsMA/R4 | 0.323 (0.193, 0.538) | **<0.001** | 0.393 (0.197, 0.725) | **0.003** |

*MA/R1* First quartile group of the MA/R ratio, *MA/R2* Second quartile group of the MA/R ratio, *MA/R3* Third quartile group of the MA/R ratio, *MA/R4* Fourth quartile group of the MA/R ratio, *MAP* Mean arterial pressure, *PT* Prothrombin time, *APTT* Activated partial thromboplastin time, *PCT* Procalcitonin, *IL* Interleukin, *R* Reaction time, *MA* Maximum amplitude, *CI* Confidence interval.

## Discussion

Thromboelastography (TEG) is capable of simultaneously monitoring both the quantity and functionality of coagulation factors, offering superior advantages over traditional coagulation tests in distinguishing hypo- and hypercoagulable states. In a prospective study involving septic patients, continuous monitoring of TEG parameters not only outperformed conventional coagulation tests in identifying hypocoagulable states but also predicted 28-day mortality (MA: HR = 4.29, *P* = 0.014) [14]. In this cohort of patients with ARF, TEG parameters also demonstrated statistically significant differences between the survival and non-survival groups (*P* < 0.001). However, the majority of patients with ARF had TEG parameters within the normal reference range upon ICU admission, which made it challenging for clinicians to use TEG parameters to identify coagulation abnormalities.

To better understand the coagulation status of patients, Savage [12] introduced the concept of the TEG MA/R ratio, obtained by dividing the MA by the R. This ratio effectively condenses the complex functions of coagulation factors, fibrinogen, and platelets into a single indicator. This index has demonstrated significant value in diagnosing coagulopathy and predicting mortality in both adult and pediatric trauma patients [15, 16]. Additionally, some researchers proposed a thrombodynamic ratio (TDR) based on R, α-angle, and MA, which may help predict deep vein thrombosis and mortality in sepsis

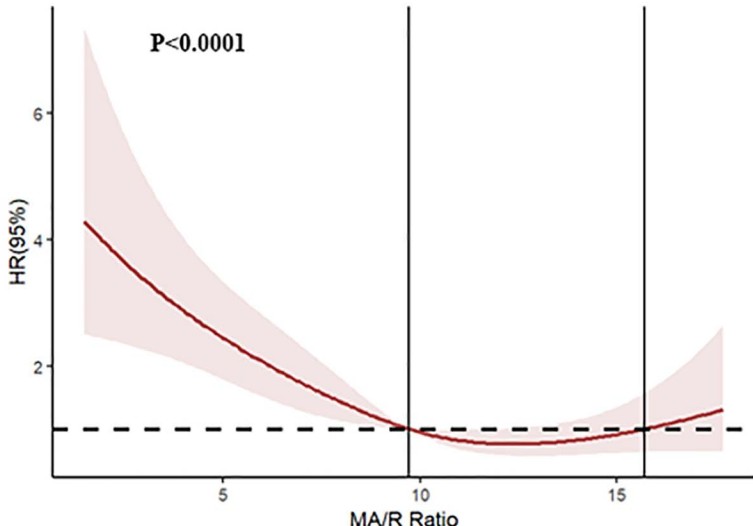

**Fig 2. Restricted cubic spline curves demonstrated a U-shaped association between the MA/R ratio and 28-day mortality.** *The colored area represents the 95% confidence interval.*

[17]. Considering the potential collinearity between K and α-angle with R and MA (VIF > 10), we categorized ARF patients into quartiles using the MA/R ratio for subgroup analysis.

Our study demonstrated a significantly elevated mortality rate in the MA/R1 group, nearly 2–3 times higher than in the other groups (P < 0.001, Table 3). Furthermore, RCS analysis revealed that lower MA/R ratios were associated with increased mortality risk (U-shaped relationship). Although mortality rates further escalated when MA/R exceeded 15.7, excessively high MA/R ratios indicated a severe hypercoagulable state, which often transitioned to a hypocoagulable state and ultimately contributed to death.

Previous research indicates that patients with ARF frequently experience dysfunction in non-pulmonary organs such as the kidneys, brain, and liver, contributing to poor prognosis [18]. Consistent with this, we observed a higher prevalence of liver and renal dysfunction, along with increased SOFA scores, in the MA/R1 group. These findings suggest that the observed association between low MA/R and poor outcomes may reflect, at least in part, the severity of the underlying primary disease and multi-organ dysfunction, rather than solely coagulation abnormalities.

Inflammation and coagulopathy represent two deeply interlinked pathways in the progression of respiratory failure. Inflammation can trigger coagulation activation, while coagulation elements such as endothelial cells and platelets reciprocally enhance inflammatory responses [19]. An imbalance in this delicate interplay predisposes patients to microvascular obstruction, alveolar-capillary leakage, and fibrin deposition along the alveolar surface, disrupting homeostasis and driving disease progression [19–21]. Choudhary et al. [22] recently conducted a large-scale analysis of sepsis-related acute respiratory failure (ARF) using a machine-learning clustering model. They identified two subgroups with multiple organ failure (MOF), both associated with high inflammatory responses. However, the MOF subgroup complicated by coagulopathy, lactic acidosis, and liver dysfunction had significantly worse outcomes than the subgroup with isolated renal and cardiac organ failure.

Laboratory data from the MA/R1 group revealed severe abnormalities in inflammatory markers, lactate levels, conventional coagulation parameters, and infection markers. Additionally, multivariate Cox regression and survival analyses confirmed that a low MA/R ratio is an independent risk factor for acute respiratory failure prognosis. Patients with low

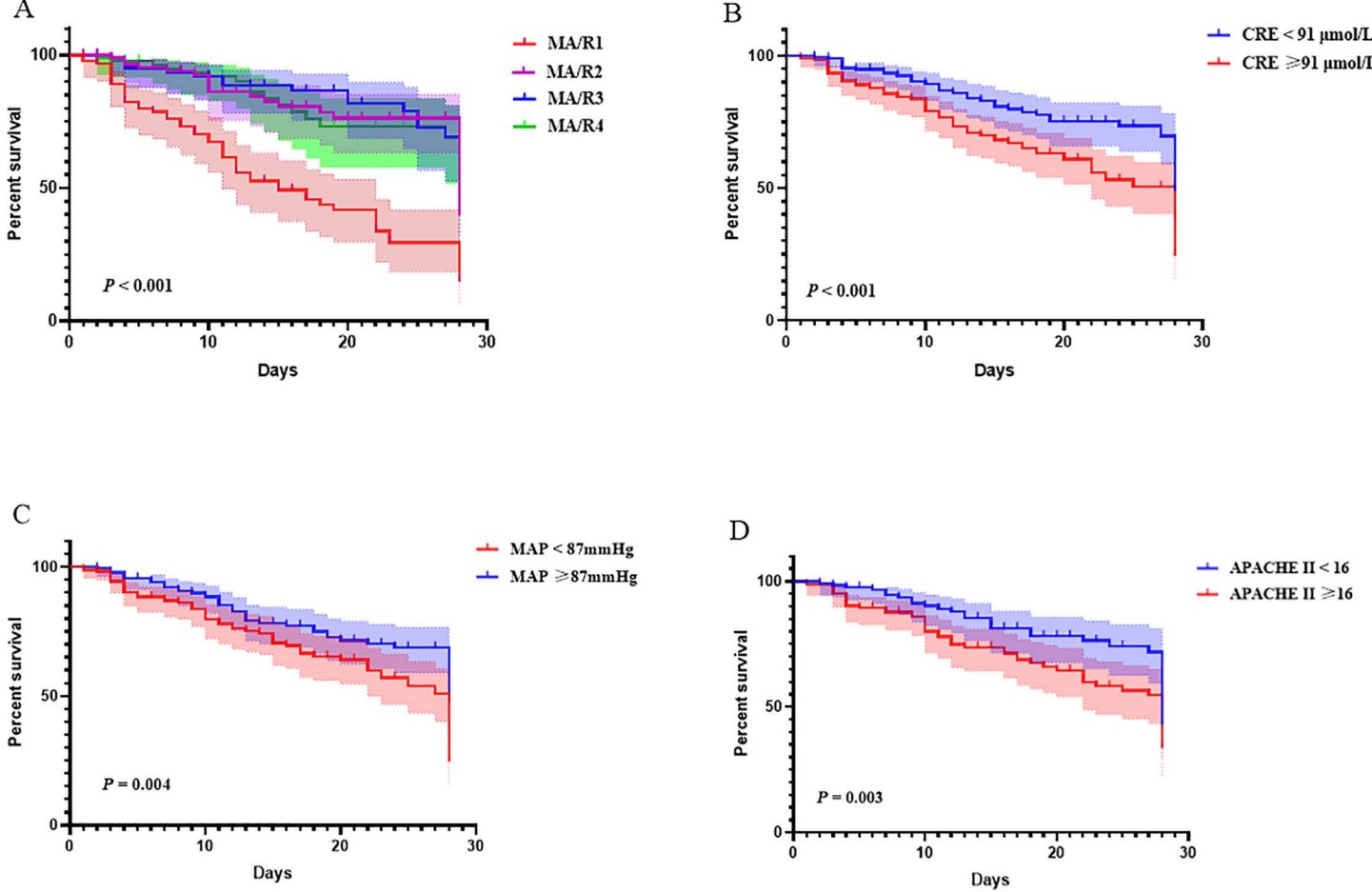

**Fig 3. Kaplan-Meier survival curves for patients based on the multivariate regression model.** *The colored area represents the 95% confidence interval.*

MA/R levels showed significantly worse outcomes. These findings highlight the essential role of coagulation, particularly the MA/R ratio, in ARF, providing a rapid and reliable tool to identify prognostic coagulation dysfunction.

To the best of our knowledge, this is the first study to evaluate the TEG MA/R ratio in acute respiratory failure, establishing a strong association between low MA/R ratios and poor prognosis. However, several limitations should be considered. Firstly, as a single-center retrospective study, the findings warrant validation through prospective, multicenter research. Secondly, comorbidities such as sepsis, diabetes, malignancies, and cardiovascular and cerebrovascular diseases may influence the results of TEG [23–25]. In this study, most comorbidities showed no significant differences across MA/R groups, except for hepatic and renal dysfunction. Future studies with larger cohorts, propensity score analyses, and adjustments for disease severity are warranted to clarify the independent contribution of MA/R to prognosis. Thirdly, some patients may have experienced COVID-19-related ARF, which is associated with a hypercoagulable state [26]. Due to the cessation of routine COVID-19 testing in late 2022, subgroup analyses by COVID-19 status were not feasible. Lastly, acute respiratory failure has multiple etiologies, such as sepsis, pneumonia, and extrapulmonary infections, which may have different effects on coagulation. The same MA/R ratio value may therefore represent different pathophysiological conditions, reducing its specificity for predicting mortality. Future studies with larger sample sizes and stratified analyses by etiology are needed to validate these findings.

In conclusion, the MA/R ratio may facilitate the early identification of coagulation abnormalities, providing an integrated measure of coagulation function and thrombus strength. Clinically, it offers simplicity and efficiency, and a low MA/R ratio appears to be an independent risk factor for mortality in acute respiratory failure. However, these findings should be interpreted with caution and require further validation through large-scale prospective and multi-center studies.

## Supporting information

**S1 Table. Thromboelastography (TEG) parameters: definitions, clinical significance, and reference ranges.** (DOCX)

**S1 Data. Anonymized clinical datasets.** (XLSX)

## Acknowledgments

Thanks to all researchers and patients for participating in this study.

## Author contributions

**Conceptualization:** Zhang-Sheng Zhao, You-Li Ma.

**Data curation:** Zhen-Zhen Wang, Bin Hu.

**Formal analysis:** Zhang-Sheng Zhao, Lei Wang.

**Funding acquisition:** Zhang-Sheng Zhao, Lei Wang.

**Writing – original draft:** Zhang-Sheng Zhao, Li-Hui Qian.

**Writing – review & editing:** Zhang-Sheng Zhao, Zhen-Zhen Wang, Bin Hu.

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
