## [Decision Letter · Decision Letter 0]

16 Sep 2025

Dear Dr. Zhang-Sheng Zhao,

Thank you for submitting your manuscript to PLOS ONE. After careful consideration, we feel that it has merit but does not fully meet PLOS ONE’s publication criteria as it currently stands. Therefore, we invite you to submit a revised version of the manuscript that addresses the points raised during the review process.

**please discuss the impact of the sepsis on the current result as the bias in the result is there with positive infection and inflammation process.**
**Also, indorse the reviewers' comments as Showen and attached.**

We look forward to receiving your revised manuscript.

Kind regards,

Rehab Al-Ansari

Academic Editor

PLOS ONE

“This work was supported by the Green Kou Foundation of the Zhejiang Blood Transfusion Association (ZJB-LK-2023-006, Zhang-Sheng Zhao), the Medical and Health Science and Technology Project of Zhejiang Province (2020KY859,Lei-Wang), and the Luili Foundation of Lihuili Hospital (2022YB004,Zhang-Sheng Zhao).”

“This work was supported by the Green Kou Foundation of the Zhejiang Blood Transfusion Association (ZJB-LK-2023-006), the Medical and Health Science and Technology Project of Zhejiang Province (2020KY859), and the Luili Foundation of Lihuili Hospital (2022YB004).”

“This work was supported by the Green Kou Foundation of the Zhejiang Blood Transfusion Association (ZJB-LK-2023-006, Zhang-Sheng Zhao), the Medical and Health Science and Technology Project of Zhejiang Province (2020KY859,Lei-Wang), and the Luili Foundation of Lihuili Hospital (2022YB004,Zhang-Sheng Zhao).”

Reviewers' comments:

Reviewer's Responses to Questions

**Comments to the Author**

1. Is the manuscript technically sound, and do the data support the conclusions?

Reviewer #1: Yes

Reviewer #2: Partly

Reviewer #3: Yes

2. Has the statistical analysis been performed appropriately and rigorously?

Reviewer #1: Yes

Reviewer #2: Yes

Reviewer #3: Yes

3. Have the authors made all data underlying the findings in their manuscript fully available?

Reviewer #1: No

Reviewer #2: Yes

Reviewer #3: No

4. Is the manuscript presented in an intelligible fashion and written in standard English?

Reviewer #1: Yes

Reviewer #2: Yes

Reviewer #3: Yes

Reviewer #1: Dear Authors,

Your study is a clinically relevant, retrospective study evaluating the prognostic utility of the thromboelastography (TEG) MA/R ratio in patients with acute respiratory failure (ARF) admitted to the ICU. The manuscript has clinical significance that addresses a crucial clinical question with high applicability in intensive care and emergency settings. It is clearly written, generally acceptable. However, abstract section and particularly tables/figures require clarification/revision.

The corrections I suggested are attached as a Word file.

Reviewer #2: This retrospective study aims to evaluate the prognostic value of the Thromboelastography (TEG) MA/R ratio in predicting mortality among patients with acute respiratory failure (ARF). The study concludes that the MA/R ratio can rapidly identify coagulation dysfunction in ARF patients and offers greater clinical relevance in predicting prognosis compared to traditional markers. A low MA/R ratio is identified as an independent risk factor for mortality in ARF. Overall, the manuscript is well-written and presents valuable findings. However, several critical issues need to be addressed to enhance the quality and clarity of the manuscript.

1. Please provide a detailed statistical analysis of the MA/R ratio under the TEG parameters entry in Table 1, specifically for the non-survival and survival groups. Please clarify and analyze the findings and highlight the differences between these two groups.

2. The rationale for classifying the MA/R ratio into four categories needs to be clearly explained in both the Methods and Results sections. Please elaborate on why a four-category classification was chosen over a three-category or two-category classification.

3. There appears to be a discrepancy between the number of patients mentioned in Line 125 (445 patients) and Figure 1 (476 patients). Please verify these numbers to ensure data accuracy and consistency throughout the manuscript.

4. The Results section would benefit from a more structured presentation. You should break down the results into bullet points or subheadings to improve readability and make it easier for readers to follow the study’s findings.

5. The manuscript would be significantly strengthened by including Receiver Operating Characteristic (ROC) curves comparing the MA/R ratio with other conventional biomarkers (such as PCT, L-lactate, etc.). This analysis may provide a clearer picture of the diagnostic and predictive efficacy of the MA/R ratio in ARF patients in the ICU setting.

Reviewer #3: General comment

This manuscript retrospectively analyzed 371 patients with acute respiratory failure to determine the prognostic value of the Thromboelastography (TEG) MA/R ratio in this population. It suggests that the MA/R ratio can identify coagulation dysfunction in patients with acute respiratory failure, thereby informing clinical treatment decisions. However, based on the study methodology and results, the use of the TEG MA/R ratio as a predictor of 28-day mortality in these patients has certain limitations. Consequently, the manuscript requires major revisions before it can be considered for acceptance.

My review comments are as follows:

1�The grouping criteria for the MA/R ratio in Table 3 are not clearly defined.

2�Given the high emphasis on research reproducibility in the medical field, the unavailability of data significantly diminishes the potential impact and societal value of this study. Even if data cannot be shared due to compliance reasons, contact information (e.g., the corresponding author's email) should be provided for future researchers to request access under the same terms. This is now a fundamental standard in academic publishing.

3�In the multivariate regression model predicting outcomes, both the "APACHE II score" and "creatinine" were included as variables. However, the APACHE II score is a composite score that already incorporates "creatinine" in its calculation. Including both a variable (creatinine) and the composite score (APACHE II) that contains it in the same model introduces serious statistical multicollinearity. This means the effect of creatinine is effectively counted twice: once directly as the "creatinine" variable and once as a component of the APACHE II score. This will lead to biased model estimates, making the Hazard Ratios (HRs) and confidence intervals for both creatinine and the APACHE II score unreliable and difficult to interpret.

4�Acute respiratory failure is a clinical syndrome that can result from various etiologies. These underlying causes can have profoundly different effects on the coagulation system. The same MA/R ratio value may stem from entirely different pathophysiological states, making it unable to accurately distinguish the underlying cause. This lack of specificity weakens its predictive value for mortality. It is recommended to increase the sample size and perform analyses stratified by the different etiologies of acute respiratory failure.

5�The MA/R ratio is susceptible to influence by the patient's inherent condition and therapeutic interventions. Its value may be more associated with the severity of the underlying primary disease rather than the coagulation state itself.

6�The conclusions of the study require further validation through prospective research.

7�Coagulation is an extremely complex and dynamic process. The MA/R ratio only extracts two parameters from specific time points, potentially overlooking other crucial information. For instance, hyperfibrinolysis is associated with bleeding risk, while hypofibrinolysis is linked to hypercoagulability. Relying solely on the MA/R ratio loses these critical distinctions, which could lead to clinical misinterpretation. A comprehensive interpretation of the entire TEG tracing is far more valuable than using a single ratio.

**Do you want your identity to be public for this peer review?** For information about this choice, including consent withdrawal, please see our Privacy Policy

Reviewer #1: No

Reviewer #2: No

Reviewer #3: No

---

## [Author Response · Author response to Decision Letter 1]

26 Nov 2025

Editors #1:

Comment 1: please discuss the impact of the sepsis on the current result as the bias in the result is there with positive infection and inflammation process.

Response 1: Thank you for raising this important point. We carefully evaluated the potential impact of sepsis on our findings. Although sepsis was more common in non-survivors than survivors (47.2% vs. 32.7%, p = 0.007), the proportion of sepsis did not differ significantly across MA/R quartiles (41.3%, 37.6%, 40.7%, 30.5%; P = 0.403, Table 2). This indicates that sepsis was evenly distributed among MA/R groups and therefore did not act as a confounder in the association between MA/R and mortality. In addition, sepsis was not associated with mortality in univariate Cox regression (HR = 0.873, P = 0.458, Table 3), further suggesting that it did not bias the observed relationship between MA/R and outcomes. Lastly, we have already addressed this issue in the limitation section of the Discussion (lines 272 - 275), where we acknowledged the potential impact of sepsis severity on coagulation parameters.

Reviewer #1:

Comment 1: However, abstract section and particularly tables/figures require clarification/revision. The corrections I suggested are attached as a Word file.

Response 1: Thank you very much for your feedback. However, we were unable to locate the Word file containing the suggested corrections for the abstract and the tables/figures. I contacted the editorial office twice, but unfortunately did not receive the file or any additional information. Therefore, we revised the manuscript and resubmitted it without addressing Reviewer 1’s comments, that were mentioned to be included in the attachment. If there are any additional comments or specific corrections we should address, please kindly let us know at your earliest convenience, and we will revise the manuscript accordingly. Thank you again for your time and assistance.

Reviewer #2:

Comment 1: Please provide a detailed statistical analysis of the MA/R ratio under the TEG parameters entry in Table 1, specifically for the non-survival and survival groups. Please clarify and analyze the findings and highlight the differences between these two groups.

Response 1: Thank you for your insightful comment. We have now provided a detailed statistical comparison of the MA/R ratio between the survival and non-survival groups in patients with acute respiratory failure (ARF). As shown in Table 1, the median MA/R ratio was significantly lower in the non-survival group (8.8 [5.0 - 11.7]) than in the survival group (10.1 [8.3 - 12.1], P < 0.001). A lower MA/R ratio reflects slower clot development and weaker clot strength relative to the initial coagulation phase, suggesting a hypocoagulable or less efficient coagulation state in non-survivors. The corresponding description has been added to the revised manuscript (Results section, lines 147-158).

Comment 2: The rationale for classifying the MA/R ratio into four categories needs to be clearly explained in both the Methods and Results sections. Please elaborate on why a four-category classification was chosen over a three-category or two-category classification.

Response 2: We appreciate the reviewer’s insightful comment. In the revised manuscript, we have added a detailed explanation for the four-category classification of the MA/R ratio in both the Methods and Results sections.

Specifically, the MA/R ratio was categorized into four groups based on its quartile distribution, following the approach reported by Savage et al. (J Trauma Acute Care Surg. 2017;83:628-634). This quartile-based classification ensures balanced subgroup sizes and enables an unbiased exploration of how outcomes vary across different strata of coagulation efficiency.

We have clarified this rationale in the Methods (lines 85 - 89) and Results (lines 155 - 158) sections of the revised manuscript.

Comment 3: There appears to be a discrepancy between the number of patients mentioned in Line 125 (445 patients) and Figure 1 (476 patients). Please verify these numbers to ensure data accuracy and consistency throughout the manuscript.

Response 3: Thank you for pointing out this discrepancy. We have carefully reviewed the data and confirmed that the correct number of enrolled patients is 476, as shown in Figure 1. The number in Line 130 has been corrected from 445 to 476 to ensure consistency throughout the manuscript.

Comment 4: The Results section would benefit from a more structured presentation. You should break down the results into bullet points or subheadings to improve readability and make it easier for readers to follow the study’s findings.

Response 4: We appreciate the reviewer’s suggestion regarding the organization of the Results section. In response, we have restructured the Results to improve clarity and readability. The section is now divided into the following subheadings: 1. Baseline characteristics of the study population;2. TEG parameters between survival and non-survival groups; 3. Association between MA/R ratio and 28-day mortality; 4. Multivariate analysis of factors associated with mortality; 5. Predictive value of MA/R ratio. We believe that this structured presentation allows readers to follow the study findings more easily.

Comment 5: The manuscript would be significantly strengthened by including Receiver Operating Characteristic (ROC) curves comparing the MA/R ratio with other conventional biomarkers (such as PCT, L-lactate, etc.). This analysis may provide a clearer picture of the diagnostic and predictive efficacy of the MA/R ratio in ARF patients in the ICU setting.

Response 5: We thank the reviewer for the valuable suggestion regarding ROC analysis. We agree that comparing the MA/R ratio with conventional biomarkers (such as PCT, L-lactate) could provide additional insight into its diagnostic and predictive performance. However, it should be noted that in our cohort, the relationship between the MA/R ratio and mortality in acute respiratory failure patients follows a U-shaped curve (Fig 2), as we have analyzed using restricted cubic spline (RCS) models (Lines 192 - 194 in the revised manuscript). Due to this non-linear relationship, a conventional ROC analysis for the MA/R ratio does not perform optimally and may not fully reflect its predictive ability. If deemed helpful by the editors or reviewers, we would be pleased to provide an ROC analysis of the MA/R ratio as a supplementary figure, while acknowledging the inherent limitations related to its U-shaped relationship

Reviewer #3:

Comment 1: The grouping criteria for the MA/R ratio in Table 3 are not clearly defined.

Response 1: Thank you for your helpful comment. We have now clarified the grouping criteria for the MA/R ratio in Table 3.

Comment 2: Given the high emphasis on research reproducibility in the medical field, the unavailability of data significantly diminishes the potential impact and societal value of this study. Even if data cannot be shared due to compliance reasons, contact information (e.g., the corresponding author's email) should be provided for future researchers to request access under the same terms. This is now a fundamental standard in academic publishing.

Response 2: We thank the reviewer for highlighting the importance of data availability. In accordance with the journal’s requirements and to ensure research reproducibility, we have anonymized the study data and uploaded them.

Comment 3: In the multivariate regression model predicting outcomes, both the "APACHE II score" and "creatinine" were included as variables. However, the APACHE II score is a composite score that already incorporates "creatinine" in its calculation. Including both a variable (creatinine) and the composite score (APACHE II) that contains it in the same model introduces serious statistical multicollinearity. This means the effect of creatinine is effectively counted twice: once directly as the "creatinine" variable and once as a component of the APACHE II score. This will lead to biased model estimates, making the Hazard Ratios (HRs) and confidence intervals for both creatinine and the APACHE II score unreliable and difficult to interpret.

Response 3: Thank you for this insightful comment. We fully agree that including both a composite score (APACHE II) and one of its components (creatinine) in the same multivariate regression model could potentially lead to multicollinearity. To address this concern, we performed a formal collinearity diagnostic using the variance inflation factor (VIF) for all variables included in the multivariate analysis (Lines 174 - 175 in the revised manuscript). The results showed that only the K value and α-angle had VIF values greater than 10 and were therefore excluded from the model. In contrast, both APACHE II score and creatinine had VIF values below 5, indicating no significant multicollinearity between these variables. This may be because creatinine represents only one component of the APACHE II score, contributing a limited portion to the total score. Therefore, both variables were retained in the final model to comprehensively evaluate their independent associations with patient outcomes.

Comment 4: Acute respiratory failure is a clinical syndrome that can result from various etiologies. These underlying causes can have profoundly different effects on the coagulation system. The same MA/R ratio value may stem from entirely different pathophysiological states, making it unable to accurately distinguish the underlying cause. This lack of specificity weakens its predictive value for mortality. It is recommended to increase the sample size and perform analyses stratified by the different etiologies of acute respiratory failure.

Response 4: We sincerely appreciate this constructive comment and fully agree with the reviewer’s observation. Acute respiratory failure is indeed a heterogeneous clinical syndrome arising from multiple etiologies, each potentially exerting distinct effects on the coagulation system. We acknowledge that the same MA/R ratio value may reflect different underlying pathophysiological conditions, which could limit its specificity and predictive power for mortality. Our current study was designed as a preliminary investigation to explore the potential prognostic value of the MA/R ratio in patients with acute respiratory failure. In future studies, we plan to expand the sample size and perform stratified analyses based on the major etiological categories, such as sepsis-dominant, pneumonia-dominant, and extrapulmonary infection groups, to validate and refine our findings. In addition, we have incorporated this important point into the Limitations section (lines 272 - 276) to acknowledge that the heterogeneity of ARF etiologies may affect the specificity of the MA/R ratio and that future stratified analyses are warranted.

Comment 5: The MA/R ratio is susceptible to influence by the patient's inherent condition and therapeutic interventions. Its value may be more associated with the severity of the underlying primary disease rather than the coagulation state itself.

Response 5: We thank the reviewer for the insightful comment. We agree that the MA/R ratio can be influenced by the patient’s underlying condition and therapeutic interventions, and may partially reflect disease severity rather than coagulation status alone. To address this, we have added text in the Discussion (lines 241 - 243) highlighting this point and emphasizing that MA/R should be interpreted in the context of overall disease severity.

Comment 6: The conclusions of the study require further validation through prospective research.

Response 6: We appreciate the reviewer’s constructive suggestion. We agree that our findings require further validation through prospective, multicenter studies. Accordingly, we have revised the conclusion to use more cautious language that reflects the exploratory nature of this study and emphasizes the need for future research (lines 277- 281).

Comment 7: Coagulation is an extremely complex and dynamic process. The MA/R ratio only extracts two parameters from specific time points, potentially overlooking other crucial information. For instance, hyperfibrinolysis is associated with bleeding risk, while hypofibrinolysis is linked to hypercoagulability. Relying solely on the MA/R ratio loses these critical distinctions, which could lead to clinical misinterpretation. A comprehensive interpretation of the entire TEG tracing is far more valuable than using a single ratio.

Response 7: We sincerely appreciate the reviewer’s insightful comment and fully agree that coagulation is a highly complex and dynamic process. As noted, the MA/R ratio represents only two parameters from specific time points and may not capture other important aspects such as fibrinolytic activity. In our study, TEG was performed at the time of ICU admission, when hyperfibrinolysis was rarely observed, whereas platelet dysfunction and coagulation abnormalities were more common. Marked fibrinolytic abnormalities are typically seen in patients who have progressed to disseminated intravascular coagulation (DIC), which was uncommon at ICU admission in our cohort. Therefore, the MA/R ratio was chosen as a simplified indicator to reflect the coagulation balance at baseline. We acknowledge that relying solely on the MA/R ratio cannot replace a full interpretation of the entire TEG tracing. However, this study was designed as a preliminary investigation, and our findings demonstrated that a low MA/R ratio was strongly associated with increased mortality, suggesting its potential prognostic value. Future prospective studies incorporating dynamic TEG monitoring and fibrinolysis parameters will further clarify the interplay between coagulation and fibrinolysis in critically ill patients.

---

## [Decision Letter · Decision Letter 1]

21 Dec 2025

The Prognostic Value of Thromboelastography MA/R Ratio in Predicting Mortality in Acute Respiratory Failure Patients

PONE-D-25-29080R1

Dear Dr. Zhang-Sheng Zhao,

We’re pleased to inform you that your manuscript has been judged scientifically suitable for publication and will be formally accepted for publication once it meets all outstanding technical requirements.

Kind regards,

Rehab Al-Ansari

Academic Editor

PLOS One

Additional Editor Comments (optional):

Reviewers' comments:

Reviewer's Responses to Questions

**Comments to the Author**

Reviewer #2: All comments have been addressed

2. Is the manuscript technically sound, and do the data support the conclusions?

Reviewer #2: Yes

3. Has the statistical analysis been performed appropriately and rigorously?

Reviewer #2: Yes

4. Have the authors made all data underlying the findings in their manuscript fully available?

Reviewer #2: No

5. Is the manuscript presented in an intelligible fashion and written in standard English?

Reviewer #2: Yes

Reviewer #2: The authors have addressed most of the concerns raised in the previous rounds of review. I have carefully examined the revisions made to the manuscript and would like to acknowledge that the quality has significantly improved. Overall, my attitude is positive

**Do you want your identity to be public for this peer review?** For information about this choice, including consent withdrawal, please see our Privacy Policy

Reviewer #2: **Yes:** Yiheng Wang

---

## [Editor Report · Acceptance letter]

PONE-D-25-29080R1

PLOS One

Dear Dr. Zhao,

I'm pleased to inform you that your manuscript has been deemed suitable for publication in PLOS One. Congratulations! Your manuscript is now being handed over to our production team.

Kind regards,

on behalf of

Dr. Rehab Al-Ansari

Academic Editor

PLOS One